# Observation of spin-space quantum transport induced by an atomic quantum point contact

Koki Ono [1✉], Toshiya Higomoto[1], Yugo Saito[1], Shun Uchino[2], Yusuke Nishida [3] & Yoshiro Takahashi [1]

Quantum transport is ubiquitous in physics. So far, quantum transport between terminals has been extensively studied in solid state systems from the fundamental point of views such as the quantized conductance to the applications to quantum devices. Recent works have demonstrated a cold-atom analog of a mesoscopic conductor by engineering a narrow conducting channel with optical potentials, which opens the door for a wealth of research of atomtronics emulating mesoscopic electronic devices and beyond. Here we realize an alternative scheme of the quantum transport experiment with ytterbium atoms in a two-orbital optical lattice system. Our system consists of a multi-component Fermi gas and a localized impurity, where the current can be created in the spin space by introducing the spin-dependent interaction with the impurity. We demonstrate a rich variety of localized-impurity-induced quantum transports, which paves the way for atomtronics exploiting spin degrees of freedom.

[1] Department of Physics, Graduate School of Science, Kyoto University, Kyoto 606-8502, Japan. [2] Advanced Science Research Center, Japan Atomic Energy Agency, Tokai, Ibaraki 319-1195, Japan. [3] Department of Physics, Tokyo Institute of Technology, Ookayama, Meguro, Tokyo 152-8551, Japan. ✉email: koukiono3@yagura.scphys.kyoto-u.ac.jp

A transport measurement between terminals has played an important role, especially for solid-state systems, in the fundamental studies of the quantum systems such as the quantized conductance and the quantum many-body effect like superconductivity and the Kondo effect as well as in the applications for electronic devices[1,2]. In recent years, the quantum simulations using ultracold atomic gases, which successfully reproduced paradigmatic models of condensed matter physics[3], have extended the domain into quantum transport experiments[4,5], often called atomtronics[6]. As a specific example, by creating a mesoscopic quantum point contact (QPC) structure in real space with sophisticatedly designed optical potentials for ultracold atoms, the quantization of conductance between two terminals, expected from the Landauer–Büttiker formula[7,8], was successfully demonstrated[9]. In addition, owing to the ability of manipulating the reservoirs or terminals that possess coherent character for the ultracold atoms isolated from an environment, the effect of fermion superfluidity of the reservoirs was revealed[10].

More recently, a scheme of a quantum transport experiment that exploits the spin degrees of freedom of ultracold atoms has been proposed[11–14]. Different from the spin transport experiments with spatially separated spin distribution[15–18], this proposal considers a spatially overlapped cloud of itinerant spinful Fermi gases interacting with a localized impurity. The itinerant atom obtains a spin-dependent phase shift via an impurity scattering, resulting in the quantum transport in the synthetic dimension of spin space instead of the real space, thus evading the need for preparation of elaborated potentials for atoms. The spin degrees of freedom of the Fermi gas and the localized impurity correspond to the terminals and the QPC, respectively. Consequently, multiterminal quantum transport via a QPC can be realized by working with the multiple spin components of atoms[14]. This spin-space scheme shares with the above-mentioned real-space scheme the coherent character of terminals consisting of ultracold atoms isolated from an environment and the controllability of the interatomic interactions. In addition, since the QPC in this scheme is also an atom with internal degrees of freedom, this system provides an intriguing possibility for the study of the nonequilibrium Anderson's orthogonality catastrophe by measuring the spin coherence of the localized impurity[13].

In this work, we successfully demonstrate the spin-space quantum transport induced by an atomic QPC using ultracold ytterbium atoms of $^{173}$Yb with the nuclear spin $I = 5/2$. By utilizing the mixed dimensional experimental platform consisting of the two-orbital system with an itinerant one-dimensional (1D) repulsively interacting Fermi gas in the ground state $|g\rangle = |^1S_0\rangle$ and a resonantly interacting impurity atom in the metastable state $|e\rangle = |^3P_0\rangle$ localized in 0D[19], we elucidate fundamental properties of the transport dynamics. Our work realizes atomtronics with a spin, providing unique possibilities in the quantum simulation of quantum transport[11–14].

## EXPERIMENTAL SCHEME

Figure 1a shows the schematic illustration of the impurity-induced quantum transport. Here, we consider the system composed of a Fermi gas with two spin components, labeled as ↑ and ↓, and a localized impurity. In our experiments, the spin degrees of freedom and the impurity correspond to the magnetic sublevels in the ground state $|g\rangle$ and the atom in the metastable state $|e\rangle$, respectively. While the spin-flip process $|\uparrow\rangle \leftrightarrow |\downarrow\rangle$ is not induced under a high magnetic field due to the energy mismatch between the initial and final states, the two spin components acquire spin-dependent phase shifts due to the impurity scattering. This scattering process is expressed as $|\sigma\rangle \rightarrow e^{2i\delta_\sigma(\varepsilon)}|\sigma\rangle$, corresponding to a unitary operator $U$ shown in Fig. 1b, where

$\delta_\sigma(\varepsilon)$ represents the scattering phase shift of the atom in the $|\sigma\rangle$ state with the kinetic energy $\varepsilon$. The scattering process of the atom in the superposition state $|+\rangle = R_{\theta=\pi/2}|\uparrow\rangle = (|\uparrow\rangle + |\downarrow\rangle)\sqrt{2}$, where $R_\theta$ is the rotation operator with an angle $\theta$, can be described as follows:

$$
\begin{aligned}
|+\rangle \rightarrow |\psi\rangle = U|+\rangle &= (e^{2i\delta_\uparrow(\varepsilon)}|\uparrow\rangle + e^{2i\delta_\downarrow(\varepsilon)}|\downarrow\rangle)/\sqrt{2} \\
&= e^{i(\delta_\uparrow(\varepsilon)+\delta_\downarrow(\varepsilon))}\Big\{\cos(\delta_\uparrow(\varepsilon) - \delta_\downarrow(\varepsilon))|+\rangle \\
&\quad + i\sin(\delta_\uparrow(\varepsilon) - \delta_\downarrow(\varepsilon))|-\rangle\Big\},
\end{aligned}
\tag{1}
$$

where $|\psi\rangle$ is the spin state after the impurity scattering and $|-\rangle = (|\uparrow\rangle - |\downarrow\rangle)/\sqrt{2}$ is orthogonal to the $|+\rangle$ state. Thus, the probability to find the $|-\rangle$ state after the impurity scattering is given by

$$
|\langle -||\psi\rangle|^2 = \sin^2(\delta_\uparrow(\varepsilon) - \delta_\downarrow(\varepsilon)),
\tag{2}
$$

showing that the spin-flip process is now induced in the $|+\rangle$ and $|-\rangle$ basis and the spin-flip probability depends on the phase shift difference between the $|\uparrow\rangle$ and $|\downarrow\rangle$ states. It should be noted that spin degrees of freedom of the $|+\rangle$ and $|-\rangle$ states are associated with the spatial degrees of freedom of left and right leads in the mesoscopic transport experiment, and thus the spin-flip process is associated with the current in the spin-space two-terminal system.

As is also shown in Fig. 1a, the differential Fermi–Dirac distribution is responsible for giving rise to the current from a source ($|+\rangle$) to a drain ($|-\rangle$), quantitatively described with the Landauer–Büttiker formula

$$
I_{+\rightarrow-} = N_{\text{imp}} \int \frac{d\varepsilon}{h} \mathcal{T}_{\theta=\pi/2}(\varepsilon)\{f(\varepsilon - \mu_+) - f(\varepsilon - \mu_-)\},
\tag{3}
$$

where $\mathcal{T}_\theta(\varepsilon)$ is the transmittance, associated with the spin-flip probability, and $h$ denotes the Planck constant. Here, $N_{\text{imp}}$ represents the number of the $|e\rangle$ atoms and $f(\varepsilon - \mu)$ is the Fermi–Dirac distribution function with a chemical potential $\mu$. The transmittance in Eq. (3) in the case of 1D is expressed as follows[14]:

$$
\mathcal{T}_{\theta=\pi/2}(\varepsilon) = \sum_{l=0,1} \sin^2(\delta_{l\uparrow}(\varepsilon) - \delta_{l\downarrow}(\varepsilon)),
\tag{4}
$$

where $l = 0$ and $l = 1$ correspond to the even wave scattering and the odd wave scattering, respectively. The scattering phase shift is calculated by numerically solving the scattering problem in the quasi-$(0+1)$D system (see Supplementary Note 1). Note that the spin-space reservoirs labeled as $|\pm\rangle$ can thermalize via the collision between the atoms in the $|\pm\rangle$ states.

The fact that the quantum transport manifests itself as the spin flip suggests that the transport phenomenon can be measured by the Ramsey sequence, as is shown in the quantum-circuit description of Fig. 1b. The first $\pi/2$-pulse creates the spin superposition state and the time interval between the two pulses is responsible for the transport time during which the atoms acquire spin-dependent phase shifts, described by the unitary operator $U$. After the second $\pi/2$-pulse, the spin state after the transport time is measured in the original $|\uparrow\rangle$ and $|\downarrow\rangle$ basis. If the localized impurity in the $|e\rangle$ state is absent, the spin population should coherently oscillate with time. In the presence of the impurity in the $|e\rangle$ state, on the other hand, the oscillation signal is expected to decay in its amplitude because of the impurity scattering phase shift. Thus, the impurity atom can be regarded as a control qubit consisting of the $|g\rangle$ and $|e\rangle$ states.

Two-orbital system with the $^{173}$Yb atoms in the $|g\rangle$ and $|e\rangle$ states can provide the experimental platform for the impurity-induced quantum transport. Using the 2D-magic-wavelength optical lattice and the 1D near-resonant optical lattice, we realize

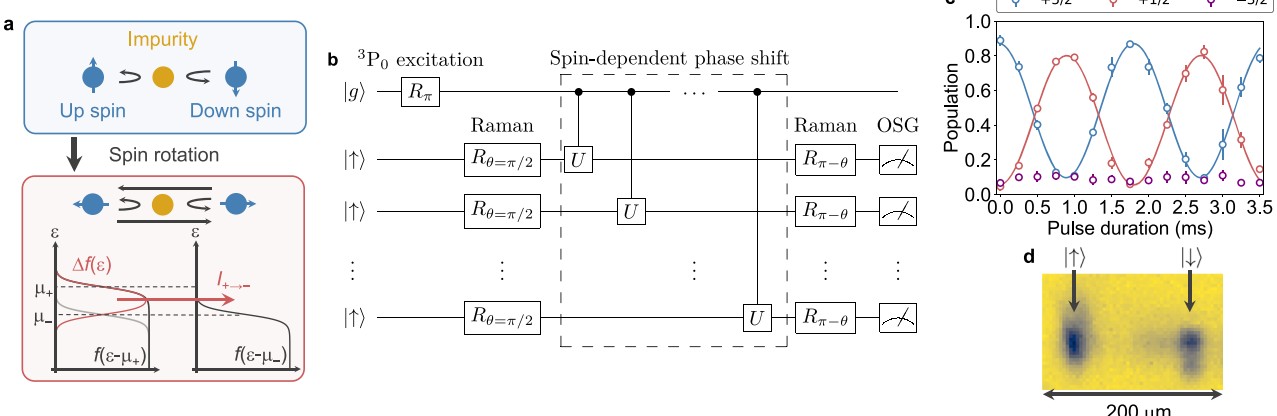

**Fig. 1 Schematic illustration of the experiment. a** Schematic representation of spin-space quantum transport. By spin rotation, quantum transport is induced in spin space. Black lines show the Fermi-Dirac distribution function, and a red line shows the differential distribution function $\Delta f(\varepsilon) = f(\varepsilon - \mu_+) - f(\varepsilon - \mu_-)$. **b** Quantum-circuit representation of a typical experimental sequence. The ${}^3P_0$ excitation ($R_\pi$) transfers the $|g\rangle|m_F = -5/2\rangle$ state to $|e\rangle|m_{F'} = -5/2\rangle$ state. The initially prepared $|\uparrow\rangle$ state is rotated by the first Raman pulse ($R_{\theta = \pi/2}$), subjected to the interaction with the impurity acquiring a spin-dependent phase shift ($U$) during the hold time, rotated again by the second Raman pulse ($R_{\pi - \theta}$), and finally detected by an OSG light. Although a single spin-flip event is shown in the circuit for simplicity, the multiple impurity scattering should occur in experiments. **c** Raman Rabi oscillation between the $|\uparrow\rangle$ and $|\downarrow\rangle$ states. Error bars show the standard deviations of the mean values obtained by averaging three measurements. Solid lines represent fits to the data. **d** Typical example of simultaneous observation of both spin states in false color time-of-flight (ToF) image of the two-component ${}^{173}Yb$ gas subjected to the OSG light. The distorted shape of the atom cloud in the $|\downarrow\rangle$ state is ascribed to the photon scattering by the OSG light.

the quasi (0 + 1)D system, where the $|g\rangle$ atom is itinerant in the 1D tube and the $|e\rangle$ atom is localized in 0D[19] (see "Methods"). In this work, the $|\uparrow\rangle$ and $|\downarrow\rangle$ states are defined as the $|+5/2\rangle = |g\rangle|m_F = +5/2\rangle$ and $|+1/2\rangle = |g\rangle|m_F = +1/2\rangle$ states, respectively, and the atom in the $|e\rangle|m_{F'} = -5/2\rangle$ state is responsible for the localized impurity, where $m_F$ denotes the projection of the hyperfine spin $F = I$ onto the quantization axis defined by a magnetic field. We perform the coherent spin manipulation using the Raman transition between the $|\uparrow\rangle$ and $|\downarrow\rangle$ states[20] (see "Methods"). Figure 1c shows the Raman Rabi oscillation between the $|\uparrow\rangle$ and $|\downarrow\rangle$ states. The interorbital interaction between the $|g\rangle$ and $|e\rangle$ atoms with the orbital Feshbach resonance[21–23] naturally realizes the spin-dependent interaction with the localized $|e\rangle$ atom. The readout of the spin state is performed by the optical Stern–Gerlach (OSG) technique[24], which enables one to separately observe the atoms in the $|\uparrow\rangle$ and $|\downarrow\rangle$ states, as shown in Fig. 1d.

## RESULTS

**Ohmic conduction**. Figure 2a shows the time evolution of the spin precession in the absence of the $|e\rangle$ atom, exhibiting the coherent oscillation of Ramsey signals with the frequency corresponding to the differential Raman light shift between $|\uparrow\rangle$ and $|\downarrow\rangle$ states (see "Methods"). As shown in Fig. 2b, on the other hand, the damping of the oscillation is observed in the presence of the $|e\rangle$ atom, indicating that the impurity-induced quantum transport is successfully demonstrated. Figure 2c represents the observed oscillation amplitude $A(t)$ as functions of the hold time. The oscillation amplitude is associated with the spin polarization, defined as $\Delta N/N$, where $\Delta N = N_+ - N_-$ and $N = N_+ + N_-$, with $N_\sigma$ being the number of atoms in the $|\sigma\rangle$ state. After the second Raman pulse of $R_{\pi - \theta}$, $N_+$ and $N_-$ correspond to $N_\uparrow$ and $N_\downarrow$, respectively. Thus, using the measured quantities of $N_\uparrow$ and $N_\downarrow$, $N_\uparrow/N = N_\uparrow/(N_\uparrow + N_\downarrow)$, and $\Delta N/N = (N_\uparrow - N_\downarrow)/(N_\uparrow + N_\downarrow)$ can be extracted. See "Methods" for the detail of the procedure of extracting $N_\uparrow/N$ and $\Delta N/N$ from the measurements. We focus on the transport dynamics after 10 ms, where $N_-$, which is the number of atoms in the drain, becomes of the order of ten and

enough to justify thermodynamic treatments[25]. The transient regime is also determined by the thermalization rate, which depends on the atom density, the scattering cross section and the atom velocity. The thermalization rate is estimated as 100 Hz, corresponding to the transient regime of $t < 10$ ms. We confirm the ohmic conduction, which manifests itself as the exponential decay of the oscillation amplitude with the finite lifetime of the $|e\rangle$ state taken into consideration (see "Methods"), and the decoherence rate is obtained as $\gamma = 47(4)$ Hz from the data fits. We note that the 25% reduction of $N$ was observed during the hold time of 40 ms, which is comparable with the $|e\rangle$ atom number loss, suggesting that the loss is caused by the inelastic collision between the $|e\rangle$ and $|g\rangle$ atoms. After 10 ms hold time, which is the temporal region of our interest, however, the reduction of $N$ is only about 8%, comparable to the uncertainty of the measurements, and thus is not taken into consideration in the analysis. The measured decoherence rate is larger than the decay rate of the $|e\rangle$ atom and smaller compared to the thermalization rate, suggesting that a quasi-steady approximation is applicable, where $N_{imp}$ and $\mu_\pm$ in Eq. (3) are replaced with those at the instantaneous time $t$. In contrast, the data at an early time deviates from the data fits after 10 ms, implying the nonlinear nature of the transport dynamics. The quantitative explanation including these data is an interesting future theory work.

More directly, we can confirm the ohmic conduction from the linearity between the current and chemical potential bias. The current $I_{+ \to -}$ which flows from $|+\rangle$ to $|-\rangle$ is defined as

$$I_{+ \to -} = -\frac{1}{2}\frac{d}{dt}\Delta N, \tag{5}$$

and thus can be extracted from the slope for $\Delta N$ in Fig. 2c. It is noted that the factor 1/2 in Eq. (5) accounts for the double counting of the decreased atom number in the $|+\rangle$ state and the increased atom number in the $|-\rangle$ state. Here $N_+$ and $N_-$ can be rewritten as functions of $\mu_+$ and $\mu_-$ based on the thermodynamics in 1D trapped fermions (see "Methods"). Circles in Fig. 2d show thus obtained chemical-potential-bias dependence of the current. Because of the finite lifetime of the $|e\rangle$ atom, the finite chemical potential bias $\Delta\mu$ is present even when the current

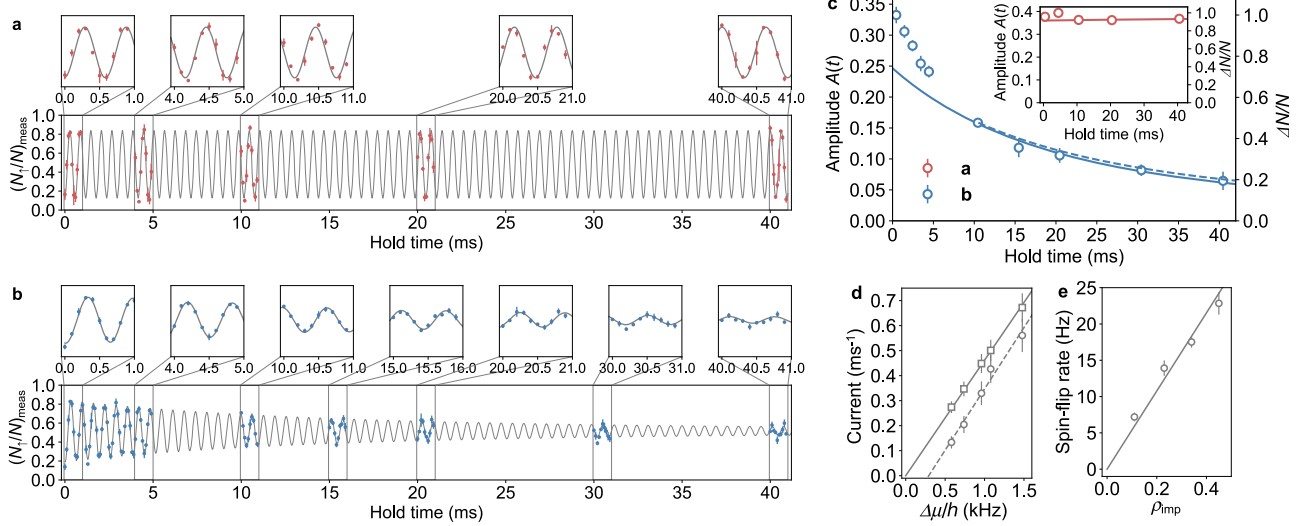

**Fig. 2 Demonstration of spin-space quantum transport induced by an atomic QPC. a, b** Time evolution of measured up-spin fraction $(N_\uparrow/N)_{\mathrm{meas}}$ at 45 Gauss: **a** in the absence of or (**b**) in the presence of the $|e\rangle$ atoms, with $\rho_{\mathrm{imp}} = 0.45$. Note that the spin-up and spin-down populations are not oscillating during the transport process before the second Raman pulse. Error bars show the standard deviations of three independent measurements. Solid lines represent guides to the eye by obtaining the fits to the data with the compensation of the magnetic field drift. Each inset shows the zoom-in view of the oscillation. **c** Time evolution of the oscillation amplitude $A(t)$ shown in (**b**). A blue solid line represents the fit to the data after 10 ms with Eq. (9), and the corresponding values of $\Delta N/N = 2N_\uparrow/N - 1 = A(t)/A(t)|_{\mathrm{max}}$ are also given on the right vertical axis (see "Methods"). A dashed line shows the numerical calculation of the transport dynamics (see Supplementary Fig. 3). The inset shows the oscillation amplitude shown in (**a**), and a red solid line shows the fit to the data with Eq. (9). Note that the normalization factor $A(t)|_{\mathrm{max}}$ is different between the blue and red data. **d** Current as a function of chemical potential bias with $\rho_{\mathrm{imp}} = 0.45$. Circles show the currents extracted from the slope of the decoherence in (**c**), and a dashed line shows the linear fit to the data. Squares are the currents obtained from $I_{+\to-} = \gamma\Delta N$ compensating for the finite impurity lifetime, and a solid line shows the linear fit to the data. In the calculation of the chemical potential, $N = 30$ is used for the total number of atoms in a tube. **e** Impurity-fraction dependence of the spin-flip rate, defined as the slope of the measured decay curve at the initial time $\frac{dA}{dt}|_{t=0} = -\gamma A_0$. Error bars in (**c**–**e**) are $1\sigma$ confidence intervals of the data fits.

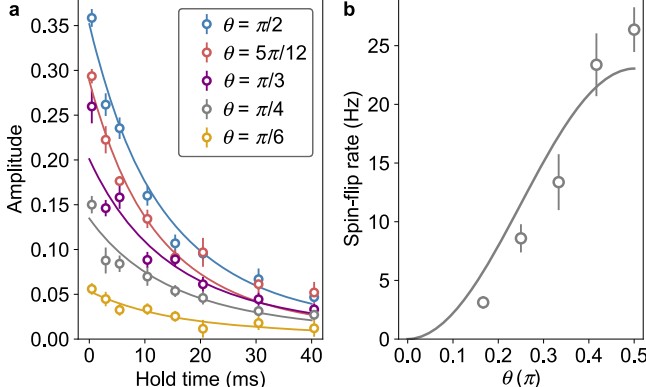

**Fig. 3 Spin-rotation-angle dependence of transport dynamics. a** Time evolution of the oscillation amplitude for different spin-rotation angles: $\theta = \pi/2, 5\pi/12, \pi/3, \pi/4,$ and $\pi/6$. Error bars are $1\sigma$ confidence intervals of the oscillation amplitude. Solid lines represent fits to the data with Eq. (9). **b** Spin-flip rate as a function of the rotation angle $\theta$. Error bars are $1\sigma$ confidence intervals of the spin-flip rate. A solid line represents a fit to the data with a sine-squared function.

asymptotically approaches to zero. In order to compensate for this finite lifetime effect, we multiply the observed current by the factor $e^{+t/\tau}$, which leads to $I_{+\to-} = \gamma\Delta N$ shown as squares in Fig. 2d. As a result, we obtain the linear dependence between the current and chemical potential bias, indicating the ohmic conduction. The conductance, defined as $I_{+\to-} = G\Delta\mu$, is obtained as $G = 0.45/h$.

We theoretically estimate the conductance from the numerical calculation of the transport dynamics, shown as a dashed line in Fig. 2c. In the calculation, we take into account the spatial

inhomogeneity of the atom numbers among many tubes, and the total atom number difference $\Delta N_{\mathrm{tot}}$ is expressed as

$$\frac{d}{dt}\Delta N_{\mathrm{tot}} = \frac{d}{dt}\sum_i \Delta N_i = -2\sum_i I_{+\to-}(\mu_{i+}, \mu_{i-}), \quad (6)$$

where $i$ means the index of the tube, and $\Delta N_i$ and $\mu_{i\pm}$ correspond to the atom number difference and the chemical potentials for the $|\pm\rangle$ states in the $i$th tube, respectively. As a result, we obtain $G = 0.41/h$, which is consistent with the measured value (see Supplementary Fig. 3).

Figure 2e shows the impurity-fraction dependence of the spin-flip rate, revealing that the spin-flip rate is proportional to $\rho_{\mathrm{imp}} = N_{\mathrm{imp}}(t=0)/N$. This is consistent with our expectation that each impurity atom serves as a single-mode QPC and the overall transport current should be proportional to the number of impurity atoms. Based on this linear dependence, the single-impurity conductance is estimated as $G_0 = 4.1 \times 10^{-2}/h$. By increasing the sensitivity of our experiment, the low impurity-fraction limit of this measurement will reveal the conductance discretized in units of $G_0$, namely in the form of $N_{\mathrm{imp}}G_0$, expected from the Landauer-Büttiker formula.

**Spin-rotation-angle dependence.** We investigate how the quantum transport dynamics depends on the choice of the basis sets of the spin states. In Fig. 2, we consider the basis set of $|+\rangle$ and $|-\rangle$ created by rotation $R_{\theta=\pi/2}$, but, in general, we can consider basis sets created by $R_\theta$ with any value of $\theta$. The generalization of the Eq. (4) for arbitrary $\theta$ is straightforward, and is given as[14]:

$$\mathcal{T}_\theta(\varepsilon) = \sin^2\theta \sum_{l=0,1} \sin^2(\delta_{l\uparrow}(\varepsilon) - \delta_{l\downarrow}(\varepsilon)), \quad (7)$$

indicating that the $\theta$ dependence of the transport current is expected to be proportional to $\sin^2\theta$. Figure 3a shows the

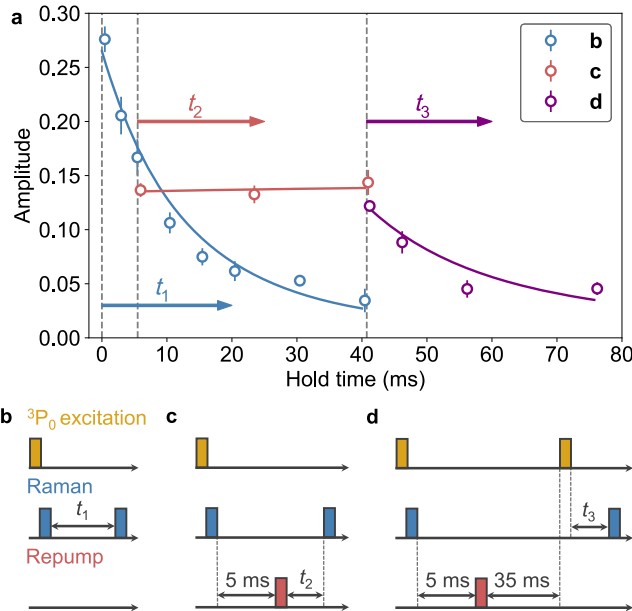

**Fig. 4 Dynamical control of transport dynamics. a** Time evolution of the oscillation amplitude for different pulse sequences. Error bars are 1$\sigma$ confidence intervals of the oscillation amplitude. Solid lines represent fits to the data with Eq. (9). Mismatches between the data points in the overlapping region of the different pulse sequences are due to experimental uncertainties. **b–d** Pulse sequences relevant to the experiments shown in (**a**).

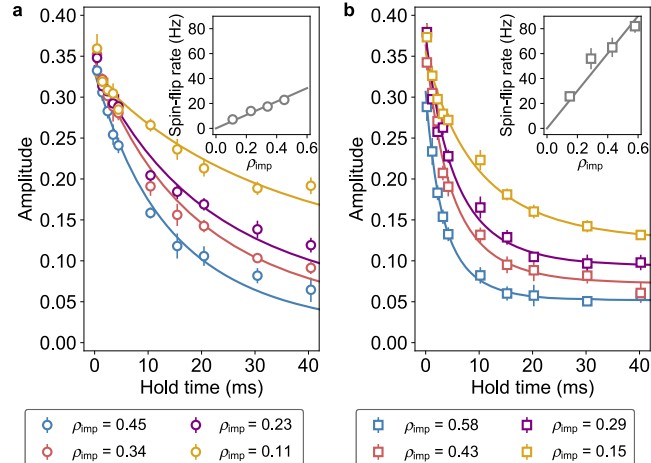

**Fig. 5 Magnetic field dependence of transport dynamics. a**, **b** Time evolution of the oscillation amplitude for different magnetic fields: **a** 45 Gauss and **b** 135 Gauss. Error bars are 1$\sigma$ confidence intervals of the oscillation amplitude. Solid lines represent the fits to the data with Eq. (9). In the data fits shown in (**b**), $\tau$ in Eq. (9) is treated as a free parameter for the better estimation of the spin-flip rate. Each inset shows the spin-flip rate as a function of the impurity fraction $\rho_{\mathrm{imp}}$. Error bars are 1$\sigma$ confidence intervals of the spin-flip rate obtained from the data fits in (**a**, **b**).

transport dynamics with different rotation angles in a magnetic field of 45 Gauss, exhibiting the clear $\theta$ dependence. Quantitatively, the spin-flip rate is obtained as the slope of the fitting curve at the initial time $t = 0$. Figure 3b shows the obtained spin-flip rate as a function of the rotation angle, which is in agreement with the expected $\sin^2 \theta$ dependence.

**Time-resolved control**. Since the QPC in this scheme is provided by individual atoms in the $|e\rangle$ state, rather than the channel structure, the quantum transport can be controlled by the excitation and de-excitation for the $|e\rangle$ atom. Figure 4a summarizes the time-resolved control of transport dynamics observed with the pulse sequences depicted in Fig. 4b–d. The experiment is performed in a magnetic field of 45 Gauss. The pulse sequence (b) illustrates the typical transport experiment similar to Fig. 2b. In the pulse sequence (c), after the transport time of 5 ms, we return the $|e\rangle$ atoms back to the $|g\rangle$ state by shining the repumping light which is resonant with the $^3P_0$–$^3D_1$ transition. This can be regarded as an operation of $\pi$-pulse in the control qubit in the quantum-circuit model of Fig. 1b. The result clearly shows the suppression of the transport. Note that the atoms should return to the ground state via several spontaneous emissions with no preferential spin components, and thus they neither contribute to the creation of spin coherence nor the decoherence. In the pulse sequence (d), after shining the repumping light we wait for 35 ms, and then apply the clock excitation pulse again to transfer the atoms in the $|-5/2\rangle$ state to the $|e\rangle$ state. As shown in Fig. 4a, the revival of the transport dynamics is observed, demonstrating the dynamical switching of the quantum transport almost at will with the excitation and de-excitation pulses.

**Control of spin-dependent interaction**. Furthermore, we investigate the possibility of controlling the spin-dependent interaction responsible for the quantum transport. Owing to the existence of the orbital Feshbach resonance between the $|g\rangle$ and $|e\rangle$ atoms of $^{173}$Yb,

the scattering phase shift depends on a magnetic field, suggesting the tunability of the transport current with a magnetic field. Figure 5a, b show the transport dynamics in a magnetic field of (a) 45 Gauss and (b) 135 Gauss for various impurity fractions $\rho_{\mathrm{imp}}$, showing that the transport current depends on the magnetic field. The result of the numerical calculation of the phase shift difference, associated with the transmittance, is consistent with the faster transport dynamics observed in a magnetic field of 135 Gauss than in 45 Gauss (see Supplementary Fig. 2). We repeat a similar transport measurement using $^{171}$Yb, which does not show an orbital Feshbach resonance in the magnetic field range of the present experiment[26]. The result shows much slow decoherence consistent with the expectation.

**Three-terminal Y-junction**. The high spin degrees of freedom of $^{173}$Yb with SU($\mathcal{N} \leq 6$) symmetry allow one to study the multi-terminal quantum transport system up to 6. Here, the SU($\mathcal{N}$) symmetry is crucial, because otherwise the spin-changing collisions take place and the system shows spin dynamics even without the localized impurities[27]. We realize the three-terminal quantum transport system by coherently connecting the $|+5/2\rangle$, $|+1/2\rangle$, and $|-3/2\rangle$ states as shown in Fig. 6a. This corresponds to the Y-junction, which has been studied theoretically[28,29] and experimentally[30,31]. We prepare the superposition with almost equal weights of the three $m_F$ states with the Raman pulse with the duration of $t_0 = 0.31$ ms (see Fig. 6b for the Raman Rabi oscillation). After the transport time, the second Raman pulse is applied with the pulse duration of $T - t_0$, and the spin population is detected in the original basis. Here $T = 1.06$ ms denotes the period of the Raman Rabi oscillation. It is noted that the remaining $|g\rangle$ atoms in the $|-5/2\rangle$ state are removed using the light resonant with the $^1S_0$–$^3P_1$($F' = 7/2$) transition since the remaining atom is undesirably coupled with the $|-1/2\rangle$ state via the Raman transition.

Figure 6c–e show the time evolution of the spin population in the presence of the $|e\rangle$ atom in a magnetic field of 119 Gauss, which clearly exhibit the decoherence of the spin precession. In the absence of the $|e\rangle$ atom, the coherent oscillation is observed with no discernible decoherence up to at least 40 ms. In Fig. 6f, the spin populations at several transport times are plotted. The

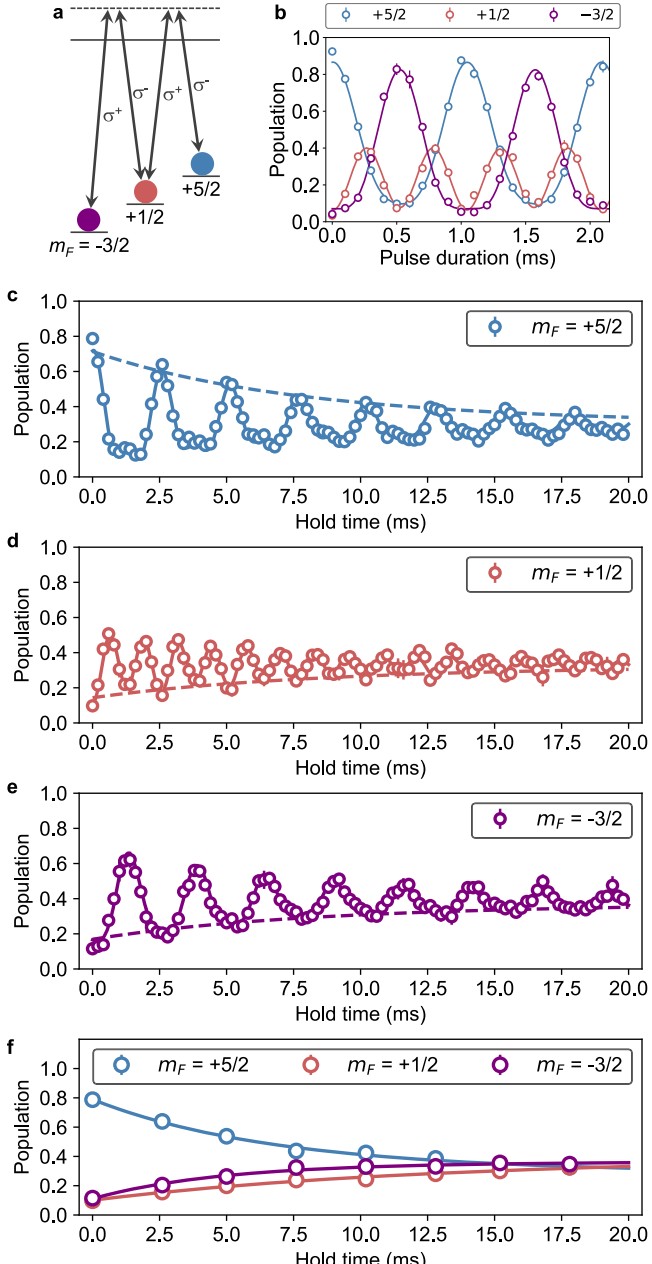

**Fig. 6 Demonstration of three-terminal quantum spin transport.**
**a** Schematic representation of the $^{173}$Yb nuclear spin states relevant to the experiment and the Raman transitions between the different $|m_F\rangle$mF states. **b** Raman Rabi oscillation of the three-level system. Error bars show the standard deviations of the mean values obtained by averaging three measurements. Solid lines represent fits to the data. **c-e** Time evolution of the relative population of the nuclear spin states: **c** $|+5/2\rangle$, **d** $|+1/2\rangle$, and **e** $|-3/2\rangle$. Error bars show the standard deviations of the mean values obtained by averaging three measurements. Solid lines represent fits to the data with the time constant fixed to 80 ms, corresponding to the lifetime of the $|e\rangle$ atom in the three-terminal experiment. During the 40 ms hold time, the reduction of the total number of the $|g\rangle$ atoms is negligible. Dashed lines show the envelopes of the data fits as guides to the eye. Time evolution of the local maxima or minima of the oscillation shown in (**c-e**). Error bars are smaller than the symbol sizes. Solid lines represent guides to the eye.

number of the terminals can be increased up to 6 by fully utilizing the spin degrees of freedom of $^{173}$Yb.

## DISCUSSION

We investigate fundamental properties of the transport dynamics such as the ohmic nature of transport and its linear dependence on the impurity atom number. We also demonstrate the controllability of the transport current via an orbital Feshbach resonance as well as the dynamical switching of the quantum transport by optical excitation of an impurity atom. In addition, the unique spin degrees of freedom of $^{173}$Yb with SU($\mathcal{N}$) symmetry enable us to successfully realize a three-terminal quantum transport system.

Our work opens up the door to the atomtronics enabled by spin degrees of freedom. Interesting future works include the nonequilibrium Anderson's orthogonality catastrophe by observing the spin dynamics of the localized $|e\rangle$ atom[11,13], the effect of interatomic interaction between $|g\rangle$ atoms[14], the full-counting statistics[13] by combining a Yb quantum-gas microscope technique[32–34], and the observation of quantized conductance. In addition, the realization of multiterminal systems with $^{173}$Yb is promising for the quantum simulation of the mesoscopic transport via a Y-junction and more complex nanostructures.

## METHODS

**Optical lattice**. A 2D array of the 1D tubes is produced using the 2D state-independent optical lattice with a wavelength of 759.4 nm. The 1D near-resonant optical lattice is superimposed along the axis of the tubes. The wavelength of the near-resonant optical lattice is chosen to be 650.7 nm, close to the $^3$P$_0$–$^3$S$_1$ transition wavelength of 649.1 nm, giving the strong confinement to the $|e\rangle$ atom alone and no net effect to the $|g\rangle$ atom[19].

After the preparation of the two-component Fermi gas ($|\pm 5/2\rangle$) with the typical atom number $4 \times 10^4$ and the temperature $0.2T_F$, where $T_F$ is the Fermi temperature, the atoms are adiabatically loaded into the optical lattices, resulting in the 2D array of about $1 \times 10^3$ 1D tubes with a typical atom number of 30 per tube. The initial lattice depths of the 2D-magic-wavelength optical lattice and the 1D near-resonant optical lattice are set to $30E_R$ and $6.8E_R$ for the $|e\rangle$ atom, respectively, where $E_R = h \times 2.0$ kHz represents the recoil energy for the magic wavelength. After the coherent transfer to the $|e\rangle$ state, the near-resonant optical lattice is ramped up to $27E_R$ to localize the $|e\rangle$ atoms. The axial and radial trap frequencies for the ground state are 76 Hz and 22 kHz, respectively, and those for the excited state 24 and 22 kHz, respectively. In this system, the $|g\rangle$ atom can be regarded as a 1D fermion since the radial vibrational energy is much larger than the Fermi energy in the central tube, which is estimated as $h \times 2$ kHz.

**Transfer to $|e\rangle$ state**. The atom in the $|g\rangle|-5/2\rangle$ state is coherently excited to the $|e\rangle|-5/2\rangle$ state with $\pi$-polarized light in a magnetic field, which is kept constant during the transport dynamics. The Rabi frequency of the clock excitation is $2\pi \times 2.5$ kHz, and the impurity fraction $\rho_{\rm imp}$ is tuned by choosing the pulse duration from 70 to 200 µs. The excitation laser is stabilized using an ultra-low-expansion glass cavity[35], and the typical linewidth is a few Hz.

**Coherent spin manipulation**. In the two-terminal experiment, the Raman light is blue-detuned by 1.00 GHz from the $^3$P$_1$ ($F' = 7/2$) state, and the polarization is perpendicular to the quantization axis defined by the magnetic field, and this results in an equal mixture of $\sigma^+$ and $\sigma^-$ polarization, realizing the Raman transition between the $|+5/2\rangle$ and $|+1/2\rangle$ states. Note that the spin-dependent light shifts induced by the Raman light alter the resonance frequency of the $|+5/2\rangle \leftrightarrow |+1/2\rangle$ Raman transition, resulting in the oscillatory behavior of the Ramsey signal, as shown in Fig. 2a, b. They also make the otherwise resonant transition to the $|-3/2\rangle$ state off-resonant and suppressed.

In the three-terminal experiment, in contrast, the Raman light is blue-detuned by 3.35 GHz from the $^3$P$_1$ ($F' = 7/2$) state, and the polarization is an equal mixture of $\pi$, $\sigma^+$, and $\sigma^-$ polarization, resulting in the equal level spacing of the $|g\rangle$ state manifold. The oscillatory behavior of the Ramsey signal shown in Fig. 6 also comes from the spin-dependent light shifts induced by the Raman light.

**Detection of spin population**. The spin population is detected with the OSG method[24], where a spin-dependent optical potential gradient is applied to separately observe the atoms in the different $|m_F\rangle$ states. The circularly polarized OSG light propagated along the quantization axis is blue-detuned by 1.13 GHz from the

$^3P_1$ ($F' = 7/2$) state. The non-negligible photon scattering associated with the OSG light results in the production of a small number of otherwise nonexisting spin components. We include the number of these spin components in the analysis, although it is typically <15%.

**Analysis of the oscillation amplitude $A(t)$.** The oscillation amplitude $A(t)$ at the hold time $t$ is obtained from the fit to the data in the time interval $t' \in [t, t + 1\,\mathrm{ms}]$ with the following function $(N_\uparrow/N)_{\mathrm{meas}} = A(t)\cos(\omega(t)t' + \varphi(t)) + B(t)$, where $\omega(t)$, $\varphi(t)$, and $B(t) \approx 1/2$ correspond to the precession angular frequency, the oscillation phase, and the offset, respectively. In addition, to compensate for the systematic effects associated with the detection process, we introduce the normalization of $2A(t)$ by its maximum value of $2A(t)|_{\max}$. We then obtain $N_\uparrow/N = 1/2 + A(t)/(2A(t)|_{\max})$.

From Eqs. (3) and (5) for the two-terminal system, the time derivative of $\Delta N$ is found to be proportional to $\Delta N$ when the current is linearized in terms of $\Delta \mu \propto \Delta N$, and the oscillation amplitude of the spin precession corresponds to $\Delta N/N$ as is mentioned in the main text. In order to analyze the transport dynamics quantitatively, the finite lifetime of the $|e\rangle$ atom $\tau$ is taken into consideration by introducing the time-dependent damping factor $e^{-t/\tau}$. Consequently, the oscillation amplitude $A(t)$ satisfies the following differential equation in terms of the transport time $t$:

$$\frac{\mathrm{d}}{\mathrm{d}t}A(t) = -\gamma e^{-t/\tau}A(t), \tag{8}$$

where $\gamma$ is associated with the decoherence rate. Solving Eq. (8) yields

$$A(t) = A_0 \exp\{-\gamma\tau(1 - e^{-t/\tau})\}. \tag{9}$$

Except for the data fits in Fig. 5b, $\tau$ is fixed to the measured lifetime of the $|e\rangle$ atom during the transport dynamics, which is 60 ms in a magnetic field of 45 Gauss.

**Thermodynamics of trapped 1D fermions.** The partition function is expressed as

$$Z = \prod_{n,\sigma}(1 + e^{\beta(\mu_\sigma - \varepsilon_n)}), \tag{10}$$

where $\varepsilon_n = \hbar\omega_{\mathrm{trap}}n$ ($n = 0, 1, 2, ...$). Here, $\omega_{\mathrm{trap}} = 2\pi \times 76$ Hz is the axial trap frequency of the tube potential. Thus, the grand potential is obtained as

$$\Omega = -\beta^{-1}\log Z = \frac{1}{\beta^2\hbar\omega_{\mathrm{trap}}}\sum_\sigma \mathrm{Li}_2(-e^{\beta\mu_\sigma}), \tag{11}$$

where $\mathrm{Li}_n(z)$ is the polylogarithm function. In this calculation, a continuous approximation $\beta\hbar\omega_{\mathrm{trap}} \to 0$ is applied to convert the sum over $n$ in $\log Z$ to an integral. Using Eq. (11), the number of the atoms in the $|\sigma\rangle$ state can be given by

$$N_\sigma = -\frac{\partial\Omega}{\partial\mu_\sigma} = \frac{1}{\beta\hbar\omega_{\mathrm{trap}}}\log(1 + e^{\beta\mu_\sigma}). \tag{12}$$

## Data availability
The datasets are available from the corresponding author on reasonable request.

## Code availability
The codes used for the numerical simulations within this paper are available from the corresponding author upon reasonable request.

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

## Acknowledgements
We thank Eunmi Chae for fruitful discussions on the multiterminal system. We also thank Shuta Namajima for careful reading of the manuscript. K.O. acknowledges support from the JSPS (KAKENHI grant number 19J11413). S.U. is supported by MEXT Leading Initiative for Excellent Young Researchers, Matsuo Foundation, and JSPS KAKENHI Grant No. JP21K03436. The experimental work was supported by the Grant-in-Aid for Scientific Research of JSPS (Nos. JP17H06138, JP18H05405, and JP18H05228), the Impulsing Paradigm Change through Disruptive Technologies (ImPACT) program, JST CREST (No. JP-MJCR1673) and MEXT Quantum Leap Flagship Program (MEXT Q-LEAP) Grant No. JPMXS0118069021.

## Author contributions
K.O., T.H., and Y.S. performed the experiment. S.U. and Y.N. performed the theoretical analysis. Y.T. supervised the whole project. All the authors discussed the results and wrote the manuscript.

## Competing interests
The authors declare no competing interests.
