## [Peer Review File · Nature Communications]

REVIEWER COMMENTS

Reviewer #1 (Remarks to the Author):

The authors investigate novel impurity-induced spin transport of ultracold atoms in effective 1D lattices. The experimental observation allows a nice explanation from the Landauer formalism and demonstrates that cold-atoms are versatile platforms for investigating quantum dynamics not easily accessible by conventional means. While the physics is important and the results are convincing, the manuscript is not publishable in its present form. Specifically,

(1) The Landauer formalism assumes the particles dump their energy when entering and leaving the system due to the differences between the reservoir chemical potentials and system energy levels. Given the good agreement between the data and the Landauer formula and the heating of the gas is slow, the authors may want to explore where the dumped energy goes. Could it be that there are collective motions or excitations of the whole cloud that take away the energy?

(2) How severe is the atom loss due to the orbital Feshbach resonance? Since a decreasing population may also decrease the amplitude, it is important to figure out its influence. Fig. 1c shows the population for a short period of time. On the scale of 40 ms, is the atom loss causing any observable effect? One may see the effect by plotting Fig. 3f to 40 ms and check if there are observable losses in the presence of the orbital Feshbach resonance.

(3) On a related note, the discussion on 3 (or more) terminal systems may distract the reader. Please consider moving it to the end and let the reader first see the full picture of spin transport.

(4) One technical note: In the analyses of data, the separation between $t > 10$ ms and $t < 10$ ms seems arbitrary. For example, it may be worth checking if the data between $10 \text{ ms} < t < 20 \text{ ms}$ differ from the steady-state value. It may also help if the authors can discuss what determines the duration of the transient regime. Is it the density, interactions, trap geometry, or a combination of those factors?

Reviewer #2 (Remarks to the Author):

In this paper, the authors experimentally demonstrate a novel implementation of quantum transport, utilizing impurity scattering to emulate a quantum point contact. The work is based on several theoretical proposals, and opens the door to future experiments on quantum transport mediated by impurity scattering. Their data on the decay of Ramsey oscillations clearly demonstrates the transport dynamics. The observation of a linear relation between the current and

chemical potential difference shows that their system provides a clean realization of transport physics.

I have a few comments and suggestions for the authors to consider.

The authors should consider citing PRL 118, 130405 (2017) from the Thywissen group and Nature Physics 9, 405–409 (2013) from the Köhl group, which also use NMR pulses to study transport in reduced dimensions.

For the data in Fig 2d, the authors should indicate the initial impurity fraction.

The definition of the "spin-flip rate" in Fig 2e is not clear. How does it relate to the decoherence rate γ ?

In the description of Fig 2a and 2b, either in the caption or the main text, it would help to specify that those are the Ramsey oscillations, so that it is clear that the spin up and down populations are not oscillating during the transport process.

The exact relationship between the spin polarization $\Delta N/N$ and the amplitude of N_{up} / N is not clear. It would help to provide a formula relating them.

The authors state that the low impurity fraction limit would reveal quantized conductance. The authors should clarify what they mean by quantized conductance in this system.

Response to Reviewers

We thank all the reviewers for careful reading of our manuscript and their constructive comments and suggestions. We have carefully revised our manuscript according to their comments.

Response to the First Reviewer:

We are very grateful to the First Reviewer for the appropriate and constructive comments. We have taken all the comments into account in the revised version of our paper. Here we describe our response to each of them.

The authors investigate novel impurity-induced spin transport of ultracold atoms in effective 1D lattices. The experimental observation allows a nice explanation from the Landauer formalism and demonstrates that cold-atoms are versatile platforms for investigating quantum dynamics not easily accessible by conventional means.

We are very grateful to the First Reviewer for his/her appreciation of this work.

While the physics is important and the results are convincing, the manuscript is not publishable in its present form. Specifically,

(1) The Landauer formalism assumes the particles dump their energy when entering and leaving the system due to the differences between the reservoir chemical potentials and system energy levels. Given the good agreement between the data and the Landauer formula and the heating of the gas is slow, the authors may want to explore where the dumped energy goes. Could it be that there are collective motions or excitations of the whole cloud that take away the energy?

Thermalization for the spin $|+\rangle$ and $|-\rangle$ states takes place with the time constant shorter than the transport time. This situation corresponds to the previous Landauer-type experiments by the ETH group [5], and our transport experiment can also be understood in a similar manner to such experiments. Figure 1 depicts the particle distribution in the two terminals of spin $|+\rangle$ and $|-\rangle$ states. In our system, the transport occurs from the initially populated spin $|+\rangle$ state, thus having a large chemical potential, to the spin $|-\rangle$ state which is initially vacant and thus has a zero chemical potential. As a result of the transport, the chemical potential difference is reduced, which should correspond to the dumping of the energy in the reviewer's comment, and no additional

energy flow into others since our system is isolated from environments.

(2) *How severe is the atom loss due to the orbital Feshbach resonance? Since a decreasing population may also decrease the amplitude, it is important to figure out its influence. Fig. 1c shows the population for a short period of time. On the scale of 40 ms, is the atom loss causing any observable effect? One may see the effect by plotting Fig. 3f to 40 ms and check if there are observable losses in the presence of the orbital Feshbach resonance.*

We thank the reviewer#1 for pointing out this effect. Let us first mention the measurements of the lifetime of the e atom. The lifetime of the e atom in the presence of g atom during the two-terminal transport dynamics is $\tau \sim 60$ ms, suggesting that the atom loss is not negligible on the scale of 40 ms.

In addition, this lifetime is shorter than the lifetime τ_0 of about 150 ms for the e atom in the absence of g atom, which is due to the inelastic collision between e and g atoms. We investigated the magnetic-field dependence of τ , and within the uncertainty of the measurement, we find no significant change in τ over a range of magnetic fields applied in our experiments.

Let us next mention the measurements of the total number N of the g atoms during the transport dynamics. Figure A shows the total number of the atoms N in Fig. 2c of the manuscript as a function of the hold time. We find that 25 % of N decreased during the hold time of 40 ms, which is comparable with the e -atom number loss. When we look at the times after 10 ms hold time, which is the temporal region of our interest in the analysis of Fig. 2c, the reduction of N is only about 8%. Since this is comparable to the error bars of the measurements of N , we did not take this into consideration in the analysis, and only consider the finite lifetime of e atom in the analysis of the oscillation amplitude $A(t)$, where the finite lifetime τ is taken into consideration by introducing the time-dependent damping factor $e^{-t/\tau}$ (see equations (8) and (9)). In other words, the effect of the finite lifetime of e atom on the transport dynamics can be compensated by multiplying $A(t)$ by $e^{+t/\tau}$.

In the revised manuscript, we add the following sentences regarding the atom loss during the transport dynamics to lines 189-197 of the revised manuscript.

“We note that we observed the 25 % reduction of N during the hold time of 40 ms, which is comparable with the $|e\rangle$ -atom number loss, suggesting that the loss is caused by the inelastic collision between the $|e\rangle$ and $|g\rangle$ atoms. After 10 ms hold time, which is the temporal region of our interest, however, the reduction of N is only about 8%, comparable to the uncertainty of the measurements, and thus is not taken into consideration in the analysis.”

FIG. A. **Total number of the atoms N in Fig. 2c as a function of the hold time.** Error bars show the standard deviation of the mean N for 1 ms.

FIG. B. **Total number of the g atoms in Fig. 3f as a function of the hold time.** Error bars show the standard deviation of the mean N for 2 ms.

In addition, the total number of the g atoms during the three-terminal transport dynamics is shown in Fig. B, and the significant change of the atom number is not found. This is consistent with the fact that the lifetime of the e -atom during the transport is measured as 80 ms, which is longer than that of the two-terminal system.

In the revised manuscript, we explicitly mentioned the negligible reduction of the total number of g atoms during 40 ms of the transport. Namely, we modified the caption of Figs. 3c-e as follows.

(Original caption)

Solid lines represent fits to the data.

(Modified caption)

Solid lines represent fits to the data with the time constant fixed to 80 ms, corresponding to the lifetime of the $|e\rangle$ atom in the three-terminal experiment. During the 40 ms hold time, the reduction of the total number of g atoms is negligible.

As a reference, Figures C(1)-(4) show the extensions of Figs 3c-f to 40 ms. It is clear that the finite oscillation amplitudes remain after 40 ms.

FIG. C. Extensions of Figs 3c-f to 40 ms. Solid lines in (1)-(3) represent the data fits with the time constant fixed to 80 ms. Solid lines in (4) represent guides to the eye.

- (3) *On a related note, the discussion on 3 (or more) terminal systems may distract the reader. Please consider moving it to the end and let the reader first see the full picture of spin transport.*

We thank the reviewer for this suggestion. We move the discussion on the three-terminal system to the last paragraph accordingly.

- (4) *One technical note: In the analyses of data, the separation between $t > 10$ ms and $t < 10$ ms seems arbitrary. For example, it may be worth checking if the data between 10 ms $< t < 20$ ms differ from the steady-state value. It may also help if the authors can discuss what determines the duration of the transient regime. Is it the density, interactions, trap geometry, or a combination of those factors?*

Please see the relevant explanation in the text L177-180 of the revised manuscript: “We focus on the transport dynamics after 10 ms, where N_- , being the number of atoms in the drain, becomes of the order of ten and enough to justify thermodynamic treatments [25].” In addition, numerical calculation shown in Fig. 2c is performed assuming a quasi-steady approximation, where N_{imp} and μ_{\pm} in equation (3) are replaced with those at the instantaneous time t . As shown in Fig. 2c, we find the qualitative agreement with the calculation and the experiment without any fitting parameters. (As is suggested by the reviewer #1, we checked that the data between 10 ms $< t < 20$ ms do not differ from the steady-state value within the uncertainty.)

The transient regime is also determined by the thermalization rate, which depends on the atom density, the scattering cross section and the atom velocity. As shown in the manuscript, the thermalization rate is estimated as 100 Hz, corresponding to the transition regime of $t < 10$ ms. We add the following sentences regarding the duration of the transient regime of the transport dynamics to lines 180-183 of the revised manuscript.

“The transient regime is also determined by the thermalization rate, which depends on the atom density, the scattering cross section and the atom velocity. The thermalization rate is estimated as 100 Hz, corresponding to the transition regime of $t < 10$ ms.”

For this addition to be consistent, the manuscript is modified as follows.

(Modified manuscript, lines 199-200)

.... compared to the thermalization rate, suggesting that ...

Response to the Second Reviewer:

We are very grateful to the Second Reviewer for the appropriate and constructive comments. We have taken all the comments into account in the revised version of our paper. Here we describe our response to each of them.

In this paper, the authors experimentally demonstrate a novel implementation of quantum transport, utilizing impurity scattering to emulate a quantum point contact. The work is based on several theoretical proposals, and opens the door to future experiments on quantum transport mediated by impurity scattering. Their data on the decay of Ramsey oscillations clearly demonstrates the transport dynamics. The observation of a linear relation between the current and chemical potential difference shows that their system provides a clean realization of transport physics.

We are very grateful to the Second Reviewer for his/her appreciation of this work.

I have a few comments and suggestions for the authors to consider.

(1) The authors should consider citing PRL 118, 130405 (2017) from the Thywissen group and Nature Physics 9, 405–409 (2013) from the Köhl group, which also use NMR pulses to study transport in reduced dimensions.

We thank the reviewer for this suggestion. We add the suggested papers in the references as follows.

(Original manuscript, line 33)

... the spin transport experiments with spatially separated spin distribution [15,16], ...

(Modified manuscript, line 33)

... the spin transport experiments with spatially separated spin distribution [15, Nature Physics 9, 405–409 (2013),16, PRL 118, 130405 (2017)], ...

(2) For the data in Fig 2d, the authors should indicate the initial impurity fraction.

We thank the reviewer for pointing out this issue. We modified the caption of Fig. 2 as follows.

(Original caption)

d. Current as a function of chemical potential bias.

(Modified caption)

d. Current as a function of chemical potential bias with $\rho_{\text{imp}} = 0.45$.

(3) *The definition of the "spin-flip rate" in Fig 2e is not clear. How does it relate to the decoherence rate gamma?*

We thank the reviewer for pointing out this issue. The spin-flip rate is defined as the slope of the measured decay curve at the initial time $\left. \frac{dA}{dt} \right|_{t=0} = -\gamma A_0$. We modified the caption of Fig. 2e as follows.

(Original caption)

Fig2 e. Impurity-fraction dependence of the spin-flip rate, defined as the slope of the measured decay curve at the initial time .

(Modified caption)

Fig2 e. Impurity-fraction dependence of the spin-flip rate, defined as the slope of the measured decay curve at the initial time. $\left. \frac{dA}{dt} \right|_{t=0} = -\gamma A_0$

(4) *In the description of Fig 2a and 2b, either in the caption or the main text, it would help to specify that those are the Ramsey oscillations, so that it is clear that the spin up and down populations are not oscillating during the transport process.*

We thank the reviewer for this suggestion. We modified the manuscript as follows.

(Original manuscript, line 164)

...exhibiting the coherent oscillation ...

(Modified manuscript, lines 161-162)

...exhibiting the coherent oscillation of Ramsey signals... .

In addition, we add the following sentence to lines 2-3 of the caption of Fig. 2a-b.

“Note that the spin up and down populations are not oscillating during the transport process before the second Raman pulse.”

- (5) *The exact relationship between the spin polarization $\Delta N/N$ and the amplitude of N_{up}/N is not clear. It would help to provide a formula relating them.*

We thank the reviewer for pointing out this issue. We add the following sentences to lines 173-177 of the revised manuscript.

“After the second Raman pulse of $R_{\pi-\theta}$, N_+ and N_- correspond to N_\uparrow and N_\downarrow , respectively. Thus, using the measured quantities of N_\uparrow and N_\downarrow , $N_\uparrow/N = N_\uparrow/(N_\uparrow + N_\downarrow)$ and $\Delta N/N = (N_\uparrow - N_\downarrow)/(N_\uparrow + N_\downarrow)$ can be extracted.”

- (6) *The authors state that the low impurity fraction limit would reveal quantized conductance. The authors should clarify what they mean by quantized conductance in this system.*

We thank the reviewer for pointing out this issue. As the localized impurity serves as a single-mode QPC, in the low impurity limit, the conductance is expected to be discretized in units of G_0 , namely in the form of $N_{\text{imp}}G_0$, expected from the Landauer-Büttiker formula. The conductance of $1/h$ will be obtained when the transmittance is unity by tuning a magnetic field in the vicinity of the orbital Feshbach resonance. We modified the manuscript as follows.

(Original manuscript, lines 236-239)

...the low impurity-fraction limit of this measurement will reveal the quantized conductance of $N_{\text{imp}}G_0$ expected from the Landauer-Büttiker formula.

(Modified manuscript, lines 249-252)

...the low impurity-fraction limit of this measurement will reveal the conductance discretized in units of G_0 , namely in the form of $N_{\text{imp}}G_0$, expected from the Landauer-Büttiker formula.

Summary of changes

General formatting

- ✓ The subsection “Three-terminal Y junction” is moved to the end of the “RESULTS” section.
- ✓ Subheadings “IMPURITY-INDUCED QUANTUM TRANSPORT” and “EXPERIMENTAL SETUP” are removed.
- ✓ The section name “CONCLUSION AND PROSPECTS” is replaced with “DISCUSSION”.
- ✓ In accordance with the guide to formatting articles, the following subscripts are changed from italic to roman.
 - $E_R \rightarrow E_R$
 - $T_F \rightarrow T_F$
- ✓ In accordance with the guide to formatting articles, equations are cited as “Equation (X)”
- ✓ In accordance with the guide to formatting articles, Symbol font is used for Greek letters in the figures.
- ✓ For consistency with the Greek letter ε in Fig. 1a, ϵ is replaced with ε .

Main text

The following line numbers refer to the revised version.

- ✓ Lines 24-25
In accordance with the guide to formatting articles, the word “new” is removed.

the novel ability \rightarrow the ability
- ✓ Line 29
In accordance with the guide to formatting articles, the word “novel” is removed.

a novel scheme of \rightarrow a scheme of

- ✓ Line 33

[15, 16] → [15, Nature Physics 9, 405–409 (2013), 16, PRL 118, 130405 (2017)]

- ✓ Line 38

In accordance with the guide to formatting articles, the following words are changed from italic to roman.

spin space → spin space

- ✓ Line 74

With the revision of the formatting, the following phrases are removed.

~~(see Experimental setup)~~

- ✓ Line 113

In accordance with the guide to formatting articles, we correct the format of the citation to the Supplementary Information.

see Supplementary Information → see Supplementary Note 1

- ✓ Line 162

...exhibiting the coherent oscillation ...

→...exhibiting the coherent oscillation of Ramsey signals... .

- ✓ Addition of the following sentences to lines 173-177

“After the second Raman pulse of $R_{\pi-\theta}$, N_+ and N_- correspond to N_\uparrow and N_\downarrow , respectively. Thus, using the measured quantities of N_\uparrow and N_\downarrow , $N_\uparrow/N = N_\uparrow/(N_\uparrow + N_\downarrow)$ and $\Delta N/N = (N_\uparrow - N_\downarrow)/(N_\uparrow + N_\downarrow)$ can be extracted.”

- ✓ Addition of the following sentences to lines 180-185

“The transient regime is also determined by the thermalization rate, which depends on the atom density, the scattering cross section and the atom velocity. The thermalization rate is estimated as 100 Hz, corresponding to the transition regime of $t < 10$ ms.”

- ✓ Addition of the following sentences to lines 191-199

“We note that the 25 % reduction of N was observed during the hold time of 40 ms, which is comparable with the $|e\rangle$ -atom number loss, suggesting that the loss is caused by the inelastic collision between the $|e\rangle$ and $|g\rangle$ atoms. After 10 ms hold time, which is the temporal region of our interest, however, the reduction of N is only about 8%, comparable to the uncertainty of the measurements, and thus is not taken into consideration in the analysis.”

- ✓ Lines 199-200

... compared to the thermalization rate ~~estimated to be~~ 100 Hz, suggesting that ...

- ✓ Lines 239-240

In accordance with the guide to formatting articles, we correct the format of the citation to the Supplementary Information.

see Supplementary Information → see Supplementary Fig. 3

- ✓ Lines 249-252

...the low impurity-fraction limit of this measurement will reveal the quantized conductance of $N_{\text{imp}}G_0$ expected from the Landauer-Büttiker formula.

→ ...the low impurity-fraction limit of this measurement will reveal the conductance discretized in units of G_0 , namely in the form of $N_{\text{imp}}G_0$, expected from the Landauer-Büttiker formula.

- ✓ Line 256

The following sentence is deleted as the subsection “Three-terminal Y-junction” is moved to the end of “RESULTS” section.

~~Hereafter, we focus on the two-terminal system.~~

- ✓ Line 298

Finally → Furthermore

- ✓ Lines 311-312

In accordance with the guide to formatting articles, we correct the format of the citation to the Supplementary Information.

see Supplementary Information → see Supplementary Fig. 2

- ✓ Lines 346-347

... the spin population ... are plotted. → ... the spin populations ... are plotted.

- ✓ Lines 353-360

As we modified the formatting of the “RESULT” section, the order of the following two sentences is changed.

The unique spin degrees of freedom ... realize a three-terminal transport system. In addition, we demonstrate ... by optical excitation of an impurity atom.

→ We also demonstrate ... by optical excitation of an impurity atom. In addition, the unique spin degrees of freedom ... realize a three-terminal transport system.

- ✓ Line 361

In accordance with the guide to formatting articles, the word “new” is removed.

opens up a new frontier in → opens up the door to

- ✓ Line 590

We add the following funding source.

SU is supported by MEXT Leading Initiative for Excellent Young Researchers and Matsuo Foundation.

→ SU is supported by MEXT Leading Initiative for Excellent Young Researchers and Matsuo Foundation, and JSPS KAKENHI Grant No. JP21K03436.

Figures

- ✓ In accordance with the guide to formatting articles, figures that use red and green simultaneously are updated with alternative color schemes.

- ✓ Caption of Fig. 1b

... the first Raman pulse($R_{\theta=\pi/2}$), interacted with the impurity...

→ ... the first Raman pulse ($R_{\theta=\pi/2}$), subjected to the interaction with the impurity ...

(Inserting a space before parentheses and changing to an appropriate phrase)

- ✓ Caption of Fig. 2a-b

Addition to the following sentence to lines 2-3 of the caption of Fig. 2a-b.

“Note that the spin up and down populations are not oscillating during the transport process before the second Raman pulse.”

- ✓ Caption of Fig. 2d

d. Current as a function of chemical potential bias.

→ **d.** Current as a function of chemical potential bias with $\rho_{\text{imp}} = 0.45$.

- ✓ Caption of Fig. 2e

e. Impurity-fraction dependence of the spin-flip rate, defined as the slope of the measured decay curve at the initial time.

→ **e.** Impurity-fraction dependence of the spin-flip rate, defined as the slope of the measured decay curve at the initial time. $\left. \frac{dA}{dt} \right|_{t=0} = -\gamma A_0$

- ✓ Caption of Figs. 3c-e

Solid lines represent fits to the data.

→ Solid lines represent fits to the data with the time constant fixed to 80 ms, corresponding to the lifetime of the $|e\rangle$ atom in the three-terminal experiment. During the 40 ms hold time, the reduction of the total number of g atoms is negligible.

- ✓ Caption of Figs. 4b

Solid lines represent fits to the data ... → Solid line represents a fit to the data...

- ✓ In accordance with the guide to formatting articles, Symbol font is used for Greek letters.

REVIEWER COMMENTS

Reviewer #1 (Remarks to the Author):

The revised manuscript has addressed my previous comments. I recommend publication after the authors address one technical issue:

It seems the red dots (case "a") on Fig. 2(c) are consistently above $(\Delta N)/N=1$. This is not possible considering the definitions of ΔN and N . Moreover, the data on Fig. 2(a) show that N_{up} is consistently smaller than N . Therefore, the problem with Fig. 2(c) may be due to a mislabeling of the values, a systematic error, a missing normalization factor, or something else.

Reviewer #2 (Remarks to the Author):

I am satisfied with the authors' responses to my comments and to those of the other referee.

Response to the First Reviewer

We are very grateful to the First Reviewer for the appropriate and constructive comment. We have taken the comment into account in the revised version of our paper. Here we describe our response:

It seems the red dots (case “a”) on Fig. 2(c) are consistently above $(\Delta N)/N=1$. This is not possible considering the definitions of ΔN and N . Moreover, the data on Fig. 2(a) show that N_{up} is consistently smaller than N . Therefore, the problem with Fig. 2(c) may be due to a mislabeling of the values, a systematic error, a missing normalization factor, or something else.

We are sorry for the insufficient explanation for this issue. This is attributed to the correction to $\Delta N/N$ due to the systematic effects such as the photon scattering by the OSG light. It is true that $\Delta N/N$ should be consistently much smaller than 1 if we adopt the quantity $(\Delta N/N)(t) \equiv 2A(t)$ that is directly obtained from the oscillation amplitude $A(t)$ in Figs. 2a and 2b. However, in the detection process of OSG, the inevitable photon scattering associated with the OSG beam should reduce $A(t)$. To compensate for this and other possible effects associated with the detection process, we made the normalization by the maximum value of the spin polarization $(\Delta N/N)_{\max}$. This is what we plotted in the original Fig. 2c with the right vertical axis as $\Delta N/N$. For the red data points in the original Fig. 2c, we adopted the same normalization “value” as the blue data points, which is the main reason of $\Delta N/N$ consistently above 1. Related to the above issue, we find the notations of N_{\uparrow}/N in Fig. 2a and 2b and $\Delta N/N$ in Fig. 2c confusing with the definition in L176 $\Delta N/N = (N_{\uparrow} - N_{\downarrow}) / (N_{\uparrow} + N_{\downarrow}) = 2N_{\uparrow}/N - 1$, which, then, should oscillate in time as shown in Figs. 2a, and 2b even though the transport dynamics is not related to the spin precession.

In the revised manuscript, the measured up-spin fraction shown in Figs 2a and 2b is denoted as $(N_{\uparrow}/N)_{\text{meas}}$, and from this oscillating quantity $(N_{\uparrow}/N)_{\text{meas}}$, we extract the oscillation amplitude $A(t)$ at the hold time t , which is obtained from the fit to the data in the time interval $t' \in [t, t + 1 \text{ ms}]$ with the following function $(N_{\uparrow}/N)_{\text{meas}}(t') = A(t) \cos(\omega(t)t' + \varphi(t)) + B(t)$, where $\omega(t)$, $\varphi(t)$, and $B(t) \approx 1/2$ correspond to the precession angular frequency, the oscillation phase, and the offset, respectively. In addition, to compensate for the systematic effects associated with the detection process, we introduce the normalization of $2A(t)$ by its maximum value $2A(t)_{\max}$. We then obtain $N_{\uparrow}/N = 1/2 + A(t)/(2A(t)_{\max})$, which is now consistent with the definition in L176 of original text.

Similar to the original Fig. 2c, the corresponding values of $\Delta N/N = 2N_{\uparrow}/N - 1 = A(t)/A(t)_{\max}$ are given on the right vertical axis in Fig. 2c. Since the normalization factor $A(t)_{\max}$ is different between the blue and red data, the two cases are separately displayed using the inset. As you can see in the inset of the revised Fig. 2c, the red data points are now consistently smaller than 1.

We thus revise Figs 2a-c and the sentence in the main text as follows.

(Original Figs. 2a and 2b)

Label of the vertical axis: N_{\uparrow}/N

Caption: Time evolution of up-spin fraction N_{\uparrow}/N at 45 Gauss:

(Revised Figs. 2a and 2b)

Label of the vertical axis: $(N_{\uparrow}/N)_{\text{meas}}$

Caption: Time evolution of measured up-spin fraction $(N_{\uparrow}/N)_{\text{meas}}$ at 45 Gauss:

(Original Fig. 2c)

Fig 2c. Time evolution of the oscillation amplitude obtained from the partial fits to the data with a sine function. Solid lines represent the fits to the data after 10 ms with Equation (9) (see Methods). The corresponding values of $\Delta N/N$ are also given on the right vertical axis. A dashed line shows the numerical calculation of the transport dynamics (see Supplementary Fig. 3).

(Revised Fig. 2c)

Fig. 2c. Time evolution of the oscillation amplitude $A(t)$ shown in Fig. 2b. A blue solid line represents the fit to the data after 10 ms with Equation (9), and the corresponding values of $\Delta N/N = 2N_{\uparrow}/N - 1 = A(t)/A(t)|_{\max}$ are also given on the right vertical axis (see Methods). A dashed line shows the numerical calculation of the transport dynamics (see Supplementary Fig. 3). The inset shows the oscillation amplitude shown in Fig. 2a, and a red solid line shows the fit to the data with Equation (9). Note that the normalization factor $A(t)|_{\max}$ is different between the blue and red data.

- (Original text L169)
oscillation amplitudes as a function of

- (Revised text L169)
oscillation amplitude $A(t)$ as a function of

- We add the following sentence to the revised text L177-178
See Methods for the detail of the procedure of extracting N_{\uparrow}/N and $\Delta N/N$ from the measurements.

- (Original text L444)
Analysis of the transport dynamics

- (Revised text L446)
Analysis of the oscillation amplitude $A(t)$

- We add the following paragraph to the revised manuscript L447-456.

“The oscillation amplitude $A(t)$ at the hold time t is obtained from the fit to the data in the time

interval $t' \in [t, t + 1 \text{ ms}]$ with the following function $(N_{\uparrow}/N)_{\text{meas}}(t') = A(t) \cos(\omega(t)t' + \varphi(t)) + B(t)$, where $\omega(t)$, $\varphi(t)$, and $B(t) \approx 1/2$ correspond to the precession angular frequency, the oscillation phase, and the offset, respectively. In addition, to compensate for the systematic effects associated with the detection process, we introduce the normalization of $2A(t)$ by its maximum value $2A(t)|_{\text{max}}$. Then, we finally obtain $N_{\uparrow}/N = 1/2 + A(t)/(2A(t)|_{\text{max}})$.”

REVIEWERS' COMMENTS

Reviewer #1 (Remarks to the Author):

The revised manuscript has addressed all my concerns. I recommend publication.